# Microstructure and Crystallization Kinetics of Silica-Based Ceramic Cores with Enhanced High-Temperature Property

**DOI:** 10.3390/ma16020606

**Published:** 2023-01-08

**Authors:** Xin Li, Shuxin Niu, Dongsheng Wang, Jie Li, Qi Jiao, Xinlong Guo, Xiqing Xu

**Affiliations:** 1Science and Technology on Advanced High Temperature Structural Materials Laboratory, Beijing Institute of Aeronautical Materials, Beijing 100095, China; 2School of Materials Science & Engineering, Chang’an University, Xi’an 710061, China

**Keywords:** ceramic cores, silica, mullite, crystallization kinetics, high-temperature property

## Abstract

Silica-based ceramic cores play key roles in the casting of aeroengine blades, but they are highly limited by the poor high-temperature mechanical property. Here, fused mullite (FM) and sintered mullite (SM) powders were modified in silica-based ceramic cores, and the microstructure evolution and crystallization kinetics of ceramic cores depending on mullite types were studied. The ceramic cores with FM showed a dense microstructure and superior mechanical properties compared to those with SM. The ceramic cores with 10 wt.% of FM showed a crystallization activation energy of 1119.5 kJ/mol and a crystallization exponent of 1.74, and the values of 938.4 kJ/mol and 1.86 as SM were employed; the decreased crystallization activation energy and the elevated crystallization exponent by SM suggested that the excess impurities of alkali oxides and alkaline-earth oxides significantly promoted the crystallization of cristobalite. Even though the ceramic cores with mullite powders decreased slightly in the room-temperature mechanical property, their high-temperature flexure strength and creep deformation resistance were enhanced. The ceramic cores with 10 wt.% of FM showed excellent comprehensive performance, with linear shrinkage of 0.69%, room-temperature strength of 18.9 MPa, and high-temperature strength of 15.5 MPa, which satisfied the demands for hollow-blade casting.

## 1. Introduction

To improve the thrust-to-weight ratio of aircraft engines, complex inner structures are designed in turbine hollow blades to provide hollow cooling paths. In the precise casting of hollow blades, ceramic cores [1,2,3] with specific shapes are generally employed to form the inner structures, which are crucial to the dimensional accuracy, qualified rate and casting cost of the resulting blades [4]. Furthermore, the ceramic cores are faced with high temperatures, complex stresses and corrosion by melt alloys in casting and solidification, which put forward serious requirements on ceramic cores, such as enough mechanical properties, low shrinkage in sintering, resistance to thermal shocks, resistance to high-temperature deformation, mechanical weakness at a certain degree to avoid crack generation in solidification, chemical stability in the superalloy liquids and high porosity to allow for chemical dissolution [5,6,7]. Therefore, ceramic cores with superior properties were crucial to precise casting technology of hollow blades and became a bottleneck in the aviation industry.

Fused silica [8,9,10,11] was a common candidate for ceramic cores, by virtue of its low thermal expansion coefficient (0.55 × 10^−6^/K between 25 °C and 1000 °C), low sintering temperatures, chemical stability against superalloy liquids and superior leachability in aqueous alkali; it was reported that 90% of the present ceramic cores were silica-based [1]. However, the resistance to high-temperature deformation required improvement, due to the low viscosity of silica glass at high temperatures. Therefore, modifiers, such as zirconia [12,13], corundum [14], silicon carbide [15] and nanograined silica [16], were generally doped in the ceramic cores so as to control the crystallization behavior of cristobalite and improve the properties of the silica-based ceramic cores.

The crystallization behavior of fused silica into cristobalite is regarded as a key factor in the properties of silica-based ceramics. According to Kazemi [17], the crystallization of cristobalite occurred at about 1380 °C on the particle surfaces of fused silica, and a large volume expansion was companied during cooling due to the α- to β-phase transition of cristobalite. In the work of Breneman [18], appropriate amounts of cristobalite in silica-based ceramic cores were beneficial to restrain the softening and shrinkage at high temperatures, and excess cristobalite would degrade the mechanical property due to the residual stress and microcracks introduced by phase transition.

Known as a high-quality refractory, mullite had excellent stability in thermal shocks, good resistance to creep deformation and high refractoriness under loads [19,20], which could act as a high-temperature stable phase in ceramics and be beneficial to the high-temperature properties of silica-based ceramic cores. However, mullite employed as mineralizer in silica-based ceramic cores was seldom reported, and relevant theoretical research was nearly vacant; therefore, the effects of mullite powders on silica-based ceramic cores and relevant crystallization kinetics in silica/mullite ceramic cores should be investigated in depth.

In this work, two types of mullite, i.e., fused mullite and sintered mullite powders, were doped in silica-based ceramic cores, and the crystallization kinetics and property evolution of ceramic cores were investigated, depending on the different mullites.

## 2. Experimental Procedure

### 2.1. Raw Materials

Fused silica powders with mean particle size of 30 μm were employed as the fundamental raw materials and two types of mullite powders as modifiers, including fused mullite and sintered mullite powders. Detailed information on the raw materials and the compositions of the mullite powders is listed in Table 1 and Table 2. Even though the two kinds of mullite powders exhibited similar purities and the particle sizes according to Table 2, the contents of Fe_2_O_3_, CaO and MgO were much higher in the sintered mullite powders, which could have a significant effect on the microstructure and properties of silica-based ceramic cores.

### 2.2. Preparation of Ceramic Cores

Each kind of mullite powder was added into the silica powders, with mass fraction of mullite powders being 0 wt.%, 5 wt.%, 10 wt.% and 15 wt.%. The ceramic cores without mullite powders were denoted as M0 in this work, the samples with 5 wt.%, 10 wt.% and 15 wt.% of fused mullite were denoted as FM5, FM10 and FM15, respectively, and the ones with sintered mullite were denoted as SM5, SM10 and SM15, respectively. After homogenous mixing, the raw powders were put into molten plasticizers, which were composed of paraffin, polyethylene and beeswax after heating and mixing at 120 °C for 12 h. After solidification of the plasticizers at room temperature, hot-pressure molding was carried out with an MPI-25t injection molding machine, to obtain green bodies of 120 mm × 10 mm × 4 mm. The green bodies were then put into a muffle furnace and sintered at 1190 °C for 6 h based on the sintering process in Figure 1. After sintering, strengthening of the ceramic cores was performed in PVA solution and TEOS solution, respectively.

### 2.3. Testing and Characterization

X-ray diffraction analysis (XRD, Cu Kα radiation, Rigaku, Tokyo, Japan) was conducted to characterize the phase composition of the ceramic cores. The microstructures in fracture surfaces were characterized using a scanning electron microscope (SEM, JSM-5600LV, JEOL, Tokyo, Japan) to investigate the morphology and fracture mechanism. Differential scanning calorimeter (DSC, STA449F3, Netzsch, Waldkraiburg, Germany) was used to explore the crystallization behavior.

The bulk density and apparent porosities were evaluated by Archimedes method [21] based on ASTM C373. Firstly, the sample was submitted to ultrasonic cleaning and drying, and the mass of the dry sample was measured as m_0_. Secondly, the sample was put into deionized water in vacuum and held for 30 min to make the water fully immerse in the porosity, and the mass of sample in water was measured as m_1_. Lastly, the sample was taken out from water, and the water on the surface was wiped using a wet cloth, after which the mass of sample with water immersed in porosity was measured as m_2_. Therefore, the bulk density and apparent porosity of samples were expressed as:(1)$$\rho =\frac{m_{0}\rho_{0}}{m_{2}-m_{1}}$$
(2)$$P_{a}=\frac{m_{2}-m_{0}}{m_{2}-m_{1}}\times 100\%$$
where ρ_0_ was the density of deionized water, m_0_ was the mass of the dry sample, m_1_ was the mass of sample weighted in water and m_2_ was the mass of sample with water immersed in porosity.

The linear shrinkage of the ceramic cores was calculated through dimensional measurement before and after sintering, using a vernier caliper. The flexural strength at room temperature was determined via three-point bending tests, using an electron-mechanical testing machine (Instron5500R, Norwood, MA, USA). High-temperature creep deformation was measured in terms of the double-cantilever beam method, after the ceramic cores were heated at 1540 °C for 30 min. Each of these values was the average of 8 measurements.

## 3. Results and Discussion

### 3.1. Phase Composition and Crystallization Kinetics

The XRD patterns of ceramic cores doped with different fractions of mullite powders were displayed in Figure 2. In the ceramic cores without mullite powders, there were only weak diffraction peaks identified as cristobalite according to PDF#27-0605 in the XRD pattern, which suggested tiny phase transformation from silica glass to cristobalite. As the ceramic cores were doped with mullite powders, diffraction peaks for mullite (PDF#15-0776) were shown in the XRD pattern, and the contents of cristobalite increased gradually with increasing fractions of mullite powders; furthermore, ceramic cores with sintered mullite showed a much higher fraction of cristobalite than those with fused mullite. Through semi-quantitative analysis [22], the mass fraction of cristobalite, mullite and amorphous silica in ceramic cores was determined depending on the XRD patterns and displayed versus the contents of mullite powders in Table 3. The ceramic cores without mullite powders showed 2.2 wt.% cristobalite, and the FM15 and SM15 samples showed cristobalite content of 14.2% and 24.0%, respectively.

As the microstructure of fused silica glass was similar to cristobalite, devitrification generally took place at high temperatures in fused silica glass and transformed to crystal cristobalite. According to Uhlmann [23], the induction period of nucleation for silica glass was up to 5 × 10^4^ s at 1300 °C based on the homogeneous nucleation theory, suggesting that the homogeneous nucleation was quite difficult at this condition. However, varieties of silica glass show devitrification at 1300 °C, suggesting inhomogeneous nucleation; moreover, crystallization is much easier in SM samples than FM samples.

To explain the unusual differences dependent on different types of mullite powders, crystallization kinetic studies were performed on the ceramic cores. DSC curves were recorded from different ceramic cores and displayed in Figure 3, with a heating rate of 10 K/min. It was confirmed that the exothermic peak at 1650.7 K (1377.5 °C) in the ceramic cores without mullite mineralizers was attributed to the crystallization of cristobalite, on the basis of XRD patterns. As 10 wt.% of mullite powders was added into the ceramic cores, the temperature of exothermic peaks was converted to 1610.5 K and 1599.4 K (1337.3 °C and 1326.2 °C) for FM10 and SM10 samples, suggesting that the mullite mineralizers were beneficial to decrease the crystallization temperature of cristobalite from silica glass. It was noted that the crystallization temperature of cristobalite in DSC curves was significantly higher than those in XRD patterns, and it was mainly attributed to the different heating programs in the two tests; the sintering temperature was maintained for 6 h before cooling and XRD analysis, whereas DSC curves were recorded without any isothermal hold.

In the crystallization of glassy materials, activation energy was demanded for the transformation from glass to crystalline phase, to overcome the energy barrier opposing the rearrangement of the constitutional units, and higher activation energy meant larger energy barrier and larger difficulty in crystallization. Therefore, the activation energy for crystallization was a key parameter for crystallization tendency in glassy materials. According to the Kissinger equation [24], the activation energy for crystallization could be calculated through
(3)$$\ln(\beta /T_{P}^{2})=-E_{c}/{\mathrm{RT}}_{p}-\ln(E_{c}/R)+\ln\nu$$
where β was the heating rate, T_P_ was the absolute temperature of exothermic peak for crystallization, E_c_ was the activation energy for crystallization, R was the gas constant and ν was a constant. By plotting $\ln(\beta /T_{P}^{2})$ against $1/T_{p}$, E_c_/R could be determined according to the slope of the straight line.

DSC curves were recorded from the ceramic cores with different heating rates, and the crystallization temperatures of exothermic peaks were listed in Table 4, and the fitting linear functions of ln(β/T_P_^2^) versus 1/T_P_ are shown in Figure 4. According to the slopes of the fitting lines, the crystallization activation energy for M0 ceramics was determined as 1318.8 kJ/mol; as 10 wt.% of fused mullite mineralizers were employed, the crystallization activation energy decreased to 1119.5 kJ/mol for FM10 samples, and the value further reduced to 938.4 kJ/mol for SM10 samples when the mineralizers were sintered mullite powders.

Another parameter to evaluate the crystallization kinetic behavior was the crystallization exponent, which could be determined according to the Augis–Bennet equation [25]
(4)$$n=\frac{2.5}{\Delta T}\times \frac{{\mathrm{RT}}_{P}^{2}}{E_{c}}$$
where ΔT was the temperature difference at the full-width half-maximum of the crystallization exothermic peaks.

The crystallization exponents were determined and are listed in Table 5, and the exponents of the ceramic cores lay between 1.5 and 2. For M0 ceramic cores, the crystallization exponent was quite close to 1.5, which meant nucleation in zero speed; in other words, once the crystal nucleus was formed in an incubation period, no new nucleus would be created, and the increased crystallinity was attributed to the nucleus growth controlled by diffusion. As the ceramic cores were doped with 10 wt.% of mullite mineralizers, the crystallization exponents for FM10 and SM10 samples increased to 1.74 and 1.86. The crystallization exponent larger than 1.5 suggested that new crystal nuclei were still forming in the period of nucleus growth, even if the nucleation speed was getting lower. On the other hand, the exponents were adjacent to 2, suggesting that the nucleus growth was mainly in two-dimensional directions.

It was noted that the fused or sintered mullite powders consisted of various impurities, including alkali oxides, alkaline-earth oxides, ferric oxide and so on. The bond strengths of Na–O, K–O, Ca–O and Mg–O were 84 kJ/mol, 54 kJ/mol, 134 kJ/mol and 154 kJ/mol, respectively. These impurities acted as a network modifier in the ceramic cores and turned to break the glass network structure in sintering, which highly promoted the transformation of silica glass to cristobalite. According to Table 2, the sintered mullite had a much higher content of alkali oxides and alkaline-earth oxides than fused mullite; therefore, the SM ceramic cores showed smaller crystallization activation energies and larger crystallization exponents than FM samples.

### 3.2. Microstructures

The linear shrinkage of the ceramic cores was measured and plotted versus the content of mullite in Figure 5. The value of linear shrinkage was 0.96 ± 0.03% in the ceramic cores with no mullite powders. As 15 wt.% of mullite powders was employed, the linear shrinkages decreased and the ceramic cores FM15 and SM15 showed linear shrinkages of 0.64 ± 0.04% and 0.33 ± 0.03%, respectively. With increasing mullite contents, the linear shrinkage was decreasing. It was suggested that, as the crystallization of cristobalite was promoted by the mullite powders, the viscosity of the ceramic cores was elevated [26], which hindered the diffusion flow of silica glass and decreased the linear shrinkages. Furthermore, the volume fraction of cristobalite was much higher in SM samples than in FM samples, which resulted in lower linear shrinkages in SM samples.

As shown in Figure 6, the apparent porosity increased with increasing contents of mullite powders. For ceramic cores without mullite powders, the apparent porosity was 26.4 ± 0.4%, with doping of mullite powders, the apparent porosity of ceramic cores FM15 and SM15 increased to 31.8 ± 0.9% and 38.6 ± 0.8%, respectively. It was obvious that the porosity was dependent on the sintering of ceramic cores and the mullite powders had larger porosity attributed to the hindering on the sintering of ceramic cores. In particular, the value of apparent porosity showed rapid growth from 32.9 ± 0.6% to 38.6 ± 0.8% as the sintered mullite content increased from 10 wt.% to 15 wt.%. The high porosities of larger than 30% were close to those in the literature [2,3,5] and were beneficial to the easy leaching of ceramic core after casting. As the fraction of crystallized cristobalite was too large, the transition of cristobalite from the α to β phase took place at 170~280 °C during cooling [18], and the phase transition was accompanied with a volume shrinkage of 2.8% and led to the rapid growth of apparent porosity.

SEM micrographs were imaged from the fracture surfaces of the different ceramic cores and are displayed in Figure 7. The ceramic cores M0 without mullite powders showed rather loose microstructure with small quantities of pores, and most of the particles displayed a transgranular fracture, as shown in Figure 7a; the fracture morphology suggested that the particles were tightly connected with each other, and the ceramic cores had high breaking energy. With the increase in fused mullite powders, the microstructures of the ceramic cores in Figure 7b,d,f became looser, which was consistent with the variation in apparent porosity in Figure 4. Due to the crystallization of cristobalite, the border of the particles became rough and exhibited quantities of small particles. As the content of fused mullite powders further increased, the sintering densification was highly hindered; therefore, larger quantities of pores between the particles were formed, and the fracture morphology gradually transformed to intergranular fracture. As the crystallization of cristobalite was easier in ceramic cores when doped with sintered mullite than fused mullite powders, the microstructures of the SM samples (Figure 7c,e,g) were much looser than the FM samples. The particle connection was loose and a large fraction of particles with intergranular fracture was shown. Moreover, the ceramic cores SM15 exhibited quantities of microcracks and chips, as marked by arrows in Figure 7g, due to the volume shrinkage of 2.8% in the α- to β-phase transition of cristobalite during cooling. As SM15 had the highest content of cristobalite with a mass fraction of 24%, the α- to β-phase transition of cristobalite led to the largest volume shrinkage and resulted in quantities of microcracks and chips in the SEM image.

### 3.3. Mechanical Properties

Bending tests were carried out at room temperature on the ceramic cores, and the flexure strength was plotted as a function of the mullite contents in Figure 8. The ceramic cores without mullite powders showed a flexure strength of up to 19.4 ± 0.52 MPa at room temperature, which is the maximum of all the ceramic cores, related to the low porosity and transgranular fracture morphology. As the fused mullite powders were added, the flexure strength gradually decreased to 18.9 ± 0.43 MPa and 17.6 ± 0.48 MPa for FM10 and FM15. The decrease in mechanical property of ceramic cores was attributed to the easier crystallization and hindered densification in FM samples and also related to the transformation from transgranular to intergranular fracture. For the SM ceramic cores, the flexure strengths were slightly lower than those for FM samples, due to the larger porosity and looser microstructures. Even though the flexure strength was 17.5 ± 0.37 MPa as the content of sintered mullite powders was 10 wt.%, the value sharply decreased to 15.3 ± 0.45 MPa for SM15 samples. It was believed that the quantities of microcracks that formed by α- to β-phase transition of cristobalite led to an obvious degradation in flexure strength.

To identify the high-temperature mechanical property of the ceramic cores, the flexure strength and creep deformation at 1540 °C were measured and plotted versus the mullite contents in Figure 9 and Figure 10. Even though the room-temperature strength was decreasing with mullite powders, the mullite powders were beneficial to the flexure strength and creep deformation resistance at 1540 °C. The ceramic cores without mullite powders showed flexure strength of 12.4 ± 0.25 MPa and creep deformation of 3.4 ± 0.42 mm. As fused mullite powders were doped in the ceramic cores, the high-temperature flexure strength increased to 15.5 ± 0.34 MPa and the creep deformation decreased to 0.3 ± 0.1 mm for FM10, which well satisfied the demands for hollow-blade casting. It was noted that mullite had excellent heat resistance compared to silica glass, which was beneficial to the mechanical property at high temperature; furthermore, the cristobalite formed in FM samples actually increased the viscosity in the ceramic cores, which further improved the high-temperature mechanical property. For FM15, the high-temperature properties declined to 13.6 ± 0.32 MPa and 0.5 ± 0.2 mm due to the larger porosity and microcracks in FM15. However, the SM samples showed degraded high-temperature mechanical properties compared to FM samples, especially for SM15 (12.5 ± 0.3 MPa and 13.1 ± 0.85 mm). On one hand, the contents of impurities, including Na_2_O, K_2_O, CaO and MgO, were much higher in the SM ceramic core; these network modifiers formed mass of phases with low melting points, which was fatal to the high-temperature mechanical property. On the other hand, the microcracks also contributed to the low strength and large creep deformation in SM ceramic cores.

## 4. Conclusions

In this work, fused mullite and sintered mullite powders were employed as modifiers in silica-based ceramic cores, and the property evolution as well as crystallization kinetics of silica-based ceramic cores were studied depending on different mullite. The fraction of cristobalite in the SM ceramic cores was significantly larger than that in FM samples, and the properties of ceramic cores are unusually different to each other, dependent on the type of mullites. In crystallization kinetic studies, the FM10 ceramic cores showed crystallization activation energy of 1119.5 kJ/mol and crystallization exponent of 1.74, and the values were 938.4 kJ/mol and 1.86 for SM10 samples, suggesting that the excess impurities of alkali oxides and alkaline-earth oxides in sintered mullite significantly promoted the crystallization of cristobalite. Even though the ceramic cores with mullite powders show a slight decrease in mechanical property at room temperature, their high temperature flexure strength and creep deformation resistance were improved; the FM10 ceramic cores showed excellent comprehensive performance, with linear shrinkage of 0.69%, room-temperature flexure strength of 18.9 MPa, high-temperature flexure strength of 15.5 MPa and creep deformation of 0.3 mm, well satisfying the demands for hollow-blade casting.

## Figures and Tables

**Figure 1 materials-16-00606-f001:**
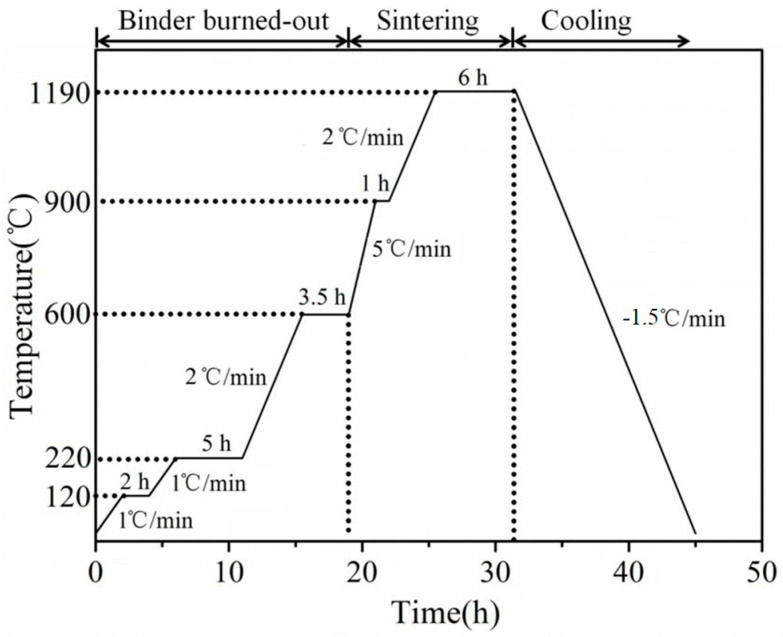
Sintering process of the ceramic cores.

**Figure 2 materials-16-00606-f002:**
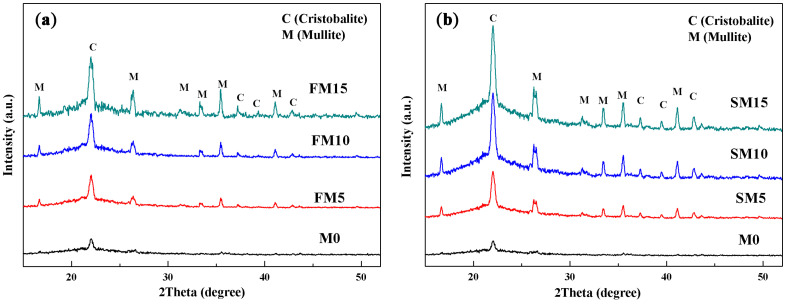
XRD patterns of the ceramic cores with different contents of (**a**) fused and (**b**) sintered mullite powders.

**Figure 3 materials-16-00606-f003:**
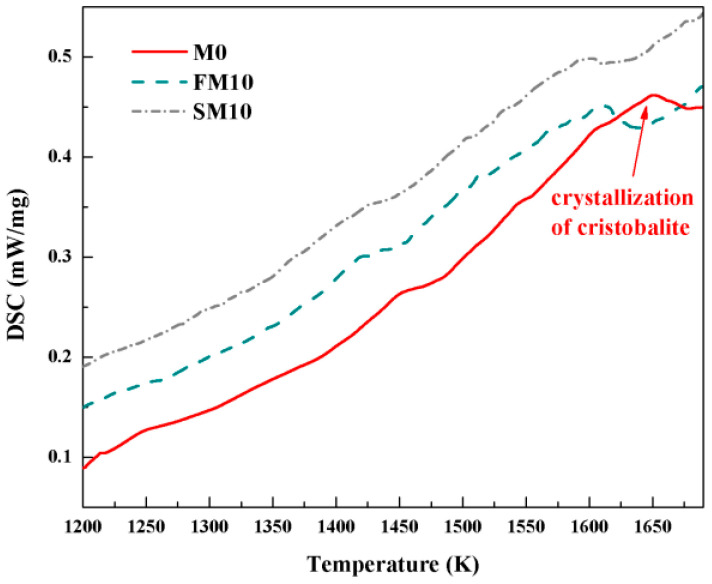
DSC curves for different ceramic cores.

**Figure 4 materials-16-00606-f004:**
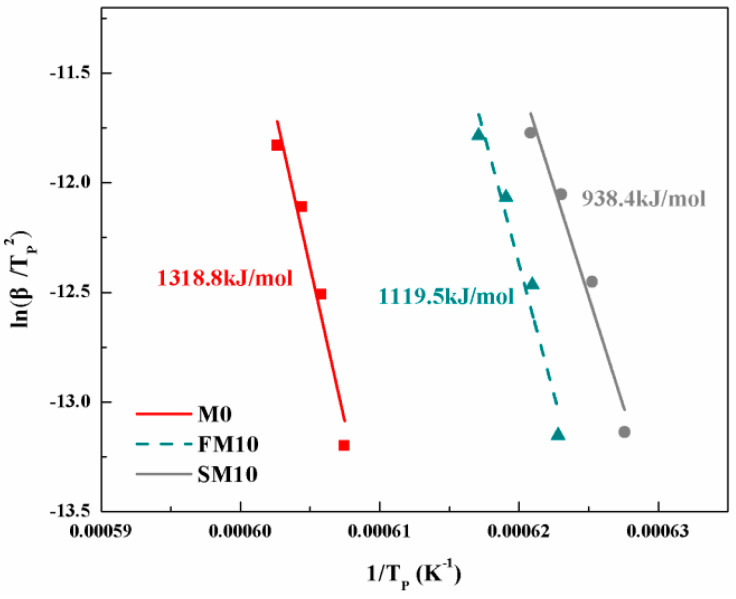
The crystallization activation energies of the different ceramic cores.

**Figure 5 materials-16-00606-f005:**
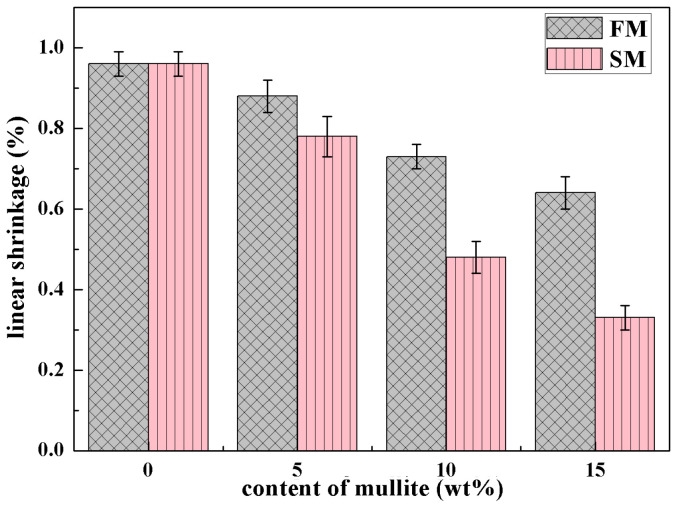
Linear shrinkage of ceramic cores with different contents of mullite.

**Figure 6 materials-16-00606-f006:**
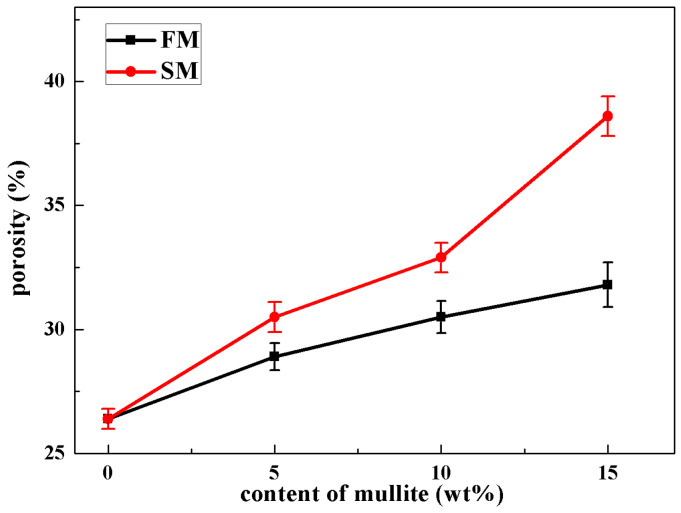
Apparent porosity of ceramic cores with different mullite contents.

**Figure 7 materials-16-00606-f007:**
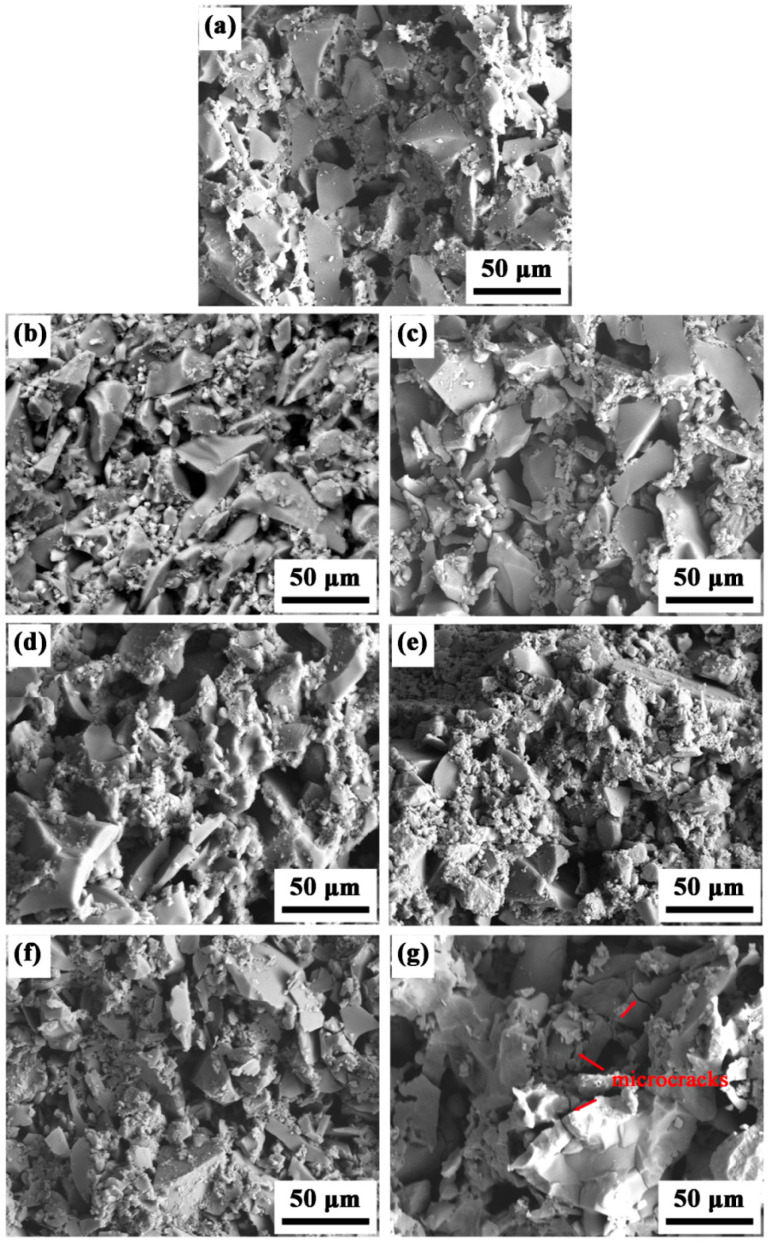
Micrographs imaged from the fracture surfaces of the different ceramic cores: (**a**) M0; (**b**) FM5; (**c**) SM5; (**d**) FM10; (**e**) SM10; (**f**) FM15; (**g**) SM15.

**Figure 8 materials-16-00606-f008:**
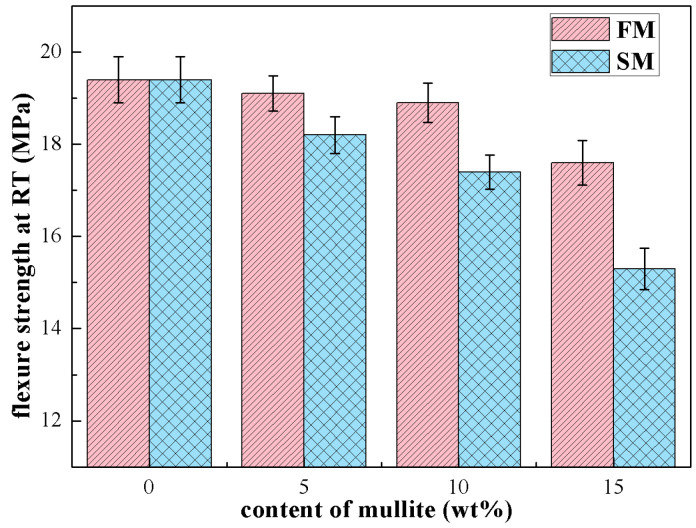
Flexure strength at room temperature of the ceramic cores with different amounts of mullite powders.

**Figure 9 materials-16-00606-f009:**
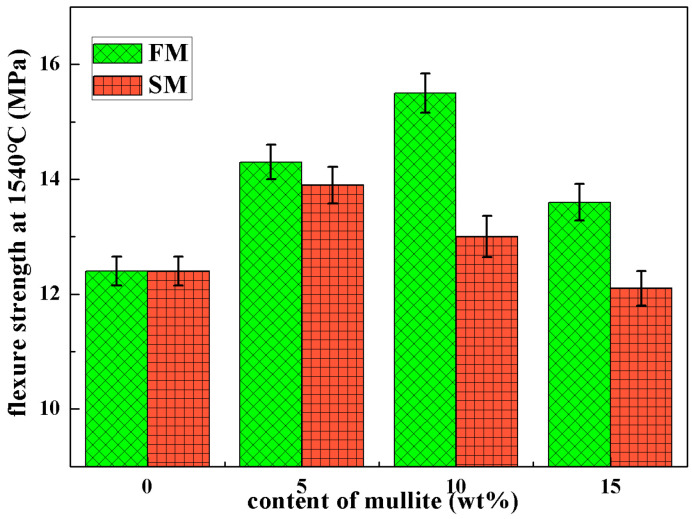
Flexure strength at 1540 °C of the ceramic cores with different amounts of mullite powders.

**Figure 10 materials-16-00606-f010:**
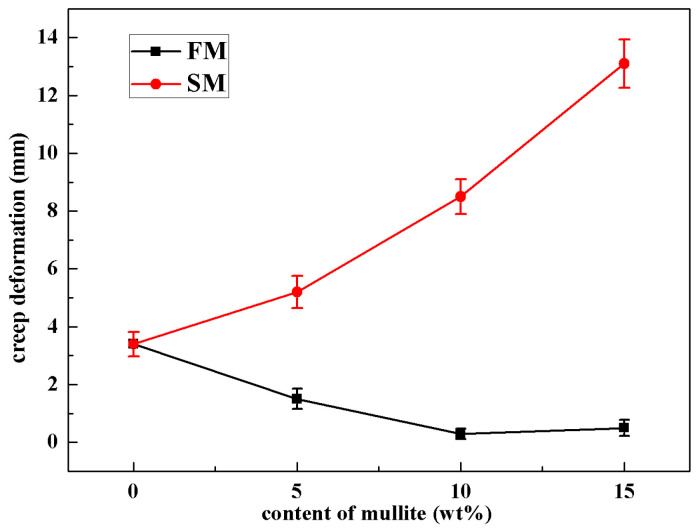
Creep deformation at 1540 °C of the ceramic cores with different amounts of mullite powders.

**Table 1 materials-16-00606-t001:** Information about raw materials.

Raw Powders	Chemical Formula	Purity	Average Grain Size	Producer
Fused silica	SiO_2_	>99.95%	30.57 μm	Jiangsu
Fused mullite	3Al_2_O_3_·2SiO_2_	>95%	18.9 μm	Henan
Sintered mullite	3Al_2_O_3_·2SiO_2_	>95%	19.9 μm	Shandong

**Table 2 materials-16-00606-t002:** Compositions of mullite powders.

Raw Powders	Compositions and Contents (wt.%)						
	Al_2_O_3_	SiO_2_	Na_2_O	K_2_O	Fe_2_O_3_	CaO	MgO
Fused mullite	75.88	22.94	0.24	0.17	0.11	0.16	<0.1
Sintered mullite	73.56	22.28	0.58	0.34	1.64	0.6	0.54

**Table 3 materials-16-00606-t003:** Mass fraction (wt.%) of the phases in different ceramic cores.

Phases	M0	FM5	FM10	FM15	SM5	SM10	SM15
cristobalite	2.2	7.9	11.5	14.2	11.4	18.9	24.1
mullite	0	5.2	10.2	15.4	5.1	10.3	15.5
amorphous silica	97.8	86.9	78.3	70.4	83.5	70.8	60.4

**Table 4 materials-16-00606-t004:** The temperatures of crystallization peak (K) in DSC curves with different heat rates.

Sample	β = 20 K/min	β = 15 K/min	β = 10 K/min	β = 5 K/min
M0	1659.4	1654.5	1650.7	1646.4
FM10	1620.5	1615.4	1610.5	1605.7
SM10	1610.8	1605.1	1599.4	1593.5

**Table 5 materials-16-00606-t005:** The crystallization exponents of the different ceramic cores.

Sample	β = 20 K/min	β = 15 K/min	β = 10 K/min	β = 5 K/min	Mean Value of n
M0	1.67	1.58	1.49	1.38	1.53
FM10	1.90	1.80	1.71	1.56	1.74
SM10	2.04	1.95	1.85	1.61	1.86

## Data Availability

Not applicable.

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
