# Peer review of "Microstructure and Crystallization Kinetics of Silica-Based Ceramic Cores with Enhanced High-Temperature Property"

_materials, 2023, doi:10.3390/ma16020606_

Round 1
Reviewer 1 Report
As a whole, the results are evaluated to be of industrially strong interest for high temperature use of silica-based ceramics with crystallized cristobalite. However, the use of pure synthetic mullite as the starting powders is desirable on the scientific standard, because the effects of the present study are attributed to the impurities included in the commercial mullite powders.
Please also refer to some grammatical comments highlighted in the text. "Crystobalite" must be also corrected as "cristobalite."

Reviewer 2 Report
Dear authors.
The article that you present is of great interest because it aims to improve the properties of silica-based ceramic cores when they are in service at high temperatures by incorporating mullite (fused and sintered). Mullite is known for its characteristics such as high resistance to thermal shock, good resistance to creep deformation, and chemical stability, among others.
Undoubtedly, by improving the high-temperature properties of the ceramic core, better performance will be obtained in the production of blades for the aviation industry.
However, the manuscript must be improved in order to be accepted for publication in the Materials journal, complying with the following points:
1.- The manuscript contains errors when mentioning the cristobalite phase, for example, crystobalite (abstract and manuscript) and crystobolite (Fig. 1).
2.- In section 2, subsection 2.2 Preparation of ceramic cores, the statement of lines 80 and 81 must be corrected, since it mentions 20 wt.% of mullite added to the ceramic cores.
3.- In lines 89 and 90, subsection 2.2 Preparation of ceramic cores, the heating rate used to prepare the ceramic body must be included.
4.- In section 2.3 Testing and characterization, the idea expressed in lines 94 to 97 is not clear.
5.- In section 3 Results and discussion, the diffraction pattern of the ceramic cores with different contents of sintered mullite (Fig. 1b), peaks of considerable intensity that were not identified can be seen, it will be important to identify them.
6.- In lines 134 to 136, it is mentioned that crystallization is easier in the SM samples compared to the FM samples, does the amount of alkali oxides influence this?
7.- In line 149 an error is detected that must be corrected.
8.- In line 164 it is mentioned that for ceramic cores without mullite content, the designation is C0, however, this does not match M0 that was handled in section 2.
9.- In the flexural resistance (Fig. 6 and 7), the bars do not show the standard deviation. On the other hand, doubts remain in the reported resistance values, for example, 17.5 MPa for the FM15 sample and 17.6 MPa for the SM10 sample, however, the bar of the FM15 sample is clearly larger.
10.- Lines 216 to 219 mention values for FM15 of 13.6 MPa and 1.2 mm, however, in Fig. 8 a value less than 1.2 mm can be seen, that is, 1.2 mm could be for sample FM5. In the same way for the SM15 sample, the reported value of 4.2 mm seems to correspond to the SM5 sample.
11.- Finally, the temperatures reported in the manuscript regarding Fig. 9 do not coincide with what the arrow indicates in the figure, this must be corrected.
Reviewer 3 Report
The article Microstructure and crystallization kinetics of silica-based ceramic cores with enhanced high-temperature property considers the evolution of the crystal structure of silicon ceramics obtained by melting. In general, this line of research is quite promising and interesting, and the presented data are of great practical importance in this area of research related to the creation of new types of ceramic materials. According to the reviewer, this article can be accepted for publication after the authors answer a number of questions that the reviewer has when analyzing this article and reading it.
1. The structure of the article should be revised, some of the results can be combined or presented in a more accurate form. For all measured quantities, it is necessary to provide data on measurement errors and standard deviations.
2. The presented X-ray diffraction patterns rather poorly reflect the changes established as a result of research when the ratio of the components changes, the authors should present comparative diffraction patterns in better quality, and also reflect a number of changes in comparison.
3. The authors should provide a comparative analysis of the ratio of the observed phases with an indication of their weight ratio with the variation of the components.
4. The authors should give more explanations on how exactly they determined the porosity, moreover, this value is quite large for ceramics.
5. The results of microimages show the presence of cracks and chips, the authors should explain what causes the formation of such defects in the structure of ceramics and how this is related to a change in the ratio of components.
6. All mechanical measurements require the presentation of the measurement error in terms of small changes.
Round 2
Reviewer 2 Report
As far as I am concerned, the authors have satisfactorily addressed all the suggestions and corrections previously requested. It is recommended to accept the manuscript for publication.
Reviewer 3 Report
The authors answered all the questions posed, the article can be accepted for publication.